# Identification and Characterization of Tomato SWI3-Like Proteins: Overexpression of *SlSWIC* Increases the Leaf Size in Transgenic *Arabidopsis*

**DOI:** 10.3390/ijms20205121

**Published:** 2019-10-16

**Authors:** Zhongyi Zhao, Tao Li, Xiuling Peng, Keqiang Wu, Songguang Yang

**Affiliations:** 1Ministry of Education Key Laboratory for Bio-Resource and Eco-Environment, College of Life Science, Sichuan University, Chengdu 610064, China; zzycwnu@163.com; 2Key Laboratory of South China Agricultural Plant Molecular Analysis and Genetic Improvement, Guangdong Provincial Key Laboratory of Applied Botany, South China Botanical Garden, Chinese Academy of Sciences, Guangzhou 510650, China; m13203206354_2@163.com; 3College of Life Sciences, China West Normal University, Nanchong 637002, China; 4Vegetable Research Institute, Guangdong Academy of Agricultural Sciences, Guangzhou 510650, China; tianxing84@163.com; 5University of Chinese Academy of Sciences, Chinese Academy of Sciences, Beijing 100049, China; 6Institute of Plant Biology, National Taiwan University, Taipei 106, Taiwan

**Keywords:** Swi3-like proteins, gene expression, protein interaction, leaf development, tomato

## Abstract

As the subunits of the SWI/SNF (mating-type switching (SWI) and sucrose nonfermenting (SNF)) chromatin-remodeling complexes (CRCs), Swi3-like proteins are crucial to chromatin remodeling in yeast and human. Growing evidence indicate that AtSWI3s are also essential for development and response to hormones in *Arabidopsis*. Nevertheless, the biological functions of Swi3-like proteins in tomato (*Solanum lycopersicum*) have not been investigated. Here we identified four Swi3-like proteins from tomato, namely SlSWI3A, SlSWI3B, SlSWI3C, and SlSWI3D. Subcellular localization analysis revealed that all SlSWI3s are localized in the nucleus. The expression patterns showed that all *SlSWI3s* are ubiquitously expressed in all tissues and organs, and *SlSWI3A* and *SlSWI3B* can be induced by cold treatment. In addition, we found that SlSWI3B can form homodimers with itself and heterodimers with SlSWI3A and SlSWI3C. SlSWI3B can also interact with SlRIN and SlCHR8, two proteins involved in tomato reproductive development. Overexpression of *SlSWI3C* increased the leaf size in transgenic *Arabidopsis* with increased expression of *GROWTH REGULATING FACTORs*, such as *GRF3*, *GRF5*, and *GRF6*. Taken together, our results indicate that SlSWI3s may play important roles in tomato growth and development.

## 1. Introduction

The fundamental unit of chromatin is the nucleosome, which is composed of two turns of DNA wrapped around a histone octamer (two H2A-H2B dimers and one H3-H4 tetramer) [1]. To readout genomic information, a dynamic chromatin environment is needed during the cellular processes such as transcription, DNA replication and recombination. The modification of chromatin can occur through multiple mechanisms, including nucleosome composition and positioning by ATP-dependent chromatin-remodeling complexes (CRCs) and enzyme complexes that modify DNA or chromatin proteins [2,3]. CRCs use energy from ATP hydrolysis [4] and change local chromatin structure, thus enriching accessibility of transcription factors and availability of genomic information [5].

The first CRC, SWI/SNF CRC, was identified in two independent screens for mutants affecting mating-type switching (SWI) and growth on sucrose (sucrose nonfermenting, SNF) in yeast [6,7]. Biochemical analysis indicated that yeast SWI/SNF CRC is composed of 12 subunits [8], and the core complex including SWI2/SNF2-type ATPase, one SNF5, and two copies of SWI3 subunits, is sufficient for execution of nucleosome remodeling in vitro [9]. Moreover, several accessory subunits associating with the core complex act as an interface for interactions with other auxiliary proteins that affect chromatin remodeling activity [10].

As part of SWI/SNF CRC, yeast Swi3p is essential for the assembly of this complex, ATP-dependent H2A-H2B displacement and recruitment to target genes [11,12]. Further genome analysis indicated that Swi3-like proteins are found in virtually all eukaryotes, such as *Drosophila*, mammals and *Arabidopsis*. Intriguingly, studies in both yeast and mammalian cells showed that Swi3p and its mammalian homologs BAF155 and BAF170 can target many genes and genomic locations in the absence of Swi2 and other components of the SWI/SNF CRC complex [13,14], suggesting that they may have unique functions in gene regulation. Generally, all Swi3-like proteins contain two typical domains, the SWIRM and SANT domains. The SWIRM domain (Swi3p, Rsc8p, and Moira) consists of 85 amino acid residues and forms a compact helix-turn-helix (HTH)-related structure [15], while the SANT (SWI3, ADA2, N-CoR and TFIIIB) domain is structurally related to the homeodomain and the Myb DNA-binding domain [16]. Furthermore, in addition to Swi3-like proteins, the SWIRM domain is also found in LSD1 (Lysine-specific demethylase 1), Ada2 (Adenosine deaminase isoenzymes 2), and in a JAB domain-containing protein involved in protein degradation through the ubiquitin pathway [15]. Functional analysis demonstrated that SANT domains tether to both DNA and proteins and are essential for histone acetyltransferase activity [16,17].

In *Arabidopsis*, four Swi3-like proteins, namely CHB1 (AtSWI3A), CHB2 (AtSWI3B), CHB3 (AtSWI3C) and CHB4 (AtSWI3D), have been identified [18,19]. Current data show that they operate as modifiers of transcriptional or epigenetic regulation in plant growth and development. For example, mutations of *AtSWI3A* and *AtSWI3B* result in disruption of embryo development at the globular stage, while mutations in *AtSWI3C* and *AtSWI3D* display severe dwarfism, abnormal vegetative development, and reduced fertility [20]. Moreover, yeast-two-hybrid assays demonstrated that AtSWI3A and AtSWI3B can form homodimers and heterodimers, and also interact with BSH/SNF5, AtSWI3C, and the flowering regulator FCA, while AtSWI3D can only bind AtSWI3B [18,20]. Interestingly, when expressed in *Saccharomyces cerevisiae*, AtSWI3B can partially complement the *swi3* mutant phenotype [18]. In addition to interacting with other SWI3 subunits, AtSWI3B also interacts with HYPERSENSITIVE TO ABA1 (HAB1), a negative regulator of ABA signaling, indicating that SWI3B is involved in the ABA signaling [21]. Subsequent research found that AtSWI3B physically interacts with IDN2, a lncRNA-binding protein, and contributes to lncRNA-mediated transcriptional silencing [22]. Recent data demonstrated that AtSWI3B (as well as AtSWI3C and AtSWI3D) interacts with MORC6 (MICRORCHIDIA 6) and a SET domain-containing histone methyltransferase SUVH9 (SU(VAR)3-9 homolog) to mediate transcriptional silencing [23]. 

In leaf development, the expression of *IAMT1* (*IAA Carboxyl Methyltransferase 1*) is regulated by AtSWI3B-mediated chromatin remodeling [24]. In addition to auxin and ABA, the *Arabidopsis* Swi3-like proteins are also involved in other hormone signaling. For instance, the *AtSWI3C* mutation down-regulates the expression of *GID1* GA receptor genes, affecting the GA perception in leaves [25]. Meanwhile, AtSWI3C physically interacts with several DELLA proteins, indicating that chromatin remodeling mediated by AtSWI3C may be required for the DELLA-mediated effects, such as activation of *GID1* [25]. Moreover, the AtSWI3C (as well as AtSWI3D)-containing SWI/SNF chromatin remodeling complex is recruited by the transcriptional coactivator AN3 (ANGUSTIFOLIA3) to the promoters of leaf development related genes [26]. Consistently, modification of the *AtSWI3C* expression increases leaf size by increasing cell number but not cell size [26]. Moreover, recent data indicated that *ZmCHB101*, one of three maize *SWI3D* genes, plays essential roles in maize growth and development at both vegetative and reproductive stages [27], since the *ZmCHB101* RNA interference plant lines displayed abaxially curling leaf and impaired tassel and cob development phenotypes.

Compared to *Arabidopsis*, the functions of Swi3-like proteins are largely unknown in other plant species. Nevertheless, phylogenetic analysis indicated that the Swi3-like proteins are widely present in the genomes of other plant species such as rice (*Oryza sativa*), *Medicago truncatula* and *Zea mays* [28]. In this study, we identified and characterized four Swi3-like proteins, namely SlSWI3A, SlSWI3B, SlSWI3C, and SlSWI3D, in tomato (*Solanum lycopersicum*). The expression profiles and subcellular localization of tomato Swi3-like proteins were investigated. In addition, the interaction of Swi3-like proteins with RIN and SlCHR8, two proteins involved in reproductive development, was also explored. Furthermore, ectopic expression of *SlSWI3C* in *Arabidopsis* resulted in the increased leaf size. Taken together, our results shed light on the potential functions of Swi3-like proteins during tomato development. 

## 2. Results

### 2.1. Identification and Phylogenetic Analysis of Tomato Swi3-Like Proteins

The Swi3-like proteins sequences of *Arabidopsis*, rice, yeast and human were used as queries to search against the SGN annotation database with the BLAST program. All sequences with an E-value below 10^−2^ were selected for further analysis. Pfam and SMART databases were used to confirm each candidate protein sequence. Like *Arabidopsis*, four Swi3-like proteins (namely SlSWI3A, SlSWI3B, SlSWI3C, and SlSWI3D) were identified in the tomato genome (Table 1). The open reading frames (ORFs) of *SlSWI3s* ranged from 1470 to 2838 bp, and the length of SlSWI3 proteins varied from 490 to 946 amino acids. Further bioinformatics analysis indicated that SlSWI3 proteins were potentially localized in different organelles including the nucleus, cytoplasm, chloroplast and mitochondria (Table 1).

To further investigate the evolutionary relationships of SlSWI3, we carried out phylogenetic analyses using Swi3-like proteins of different species. The phylogenetic tree indicated that the plant Swi3-like proteins can be clearly divided into four groups: SWI3A, SWI3B, SWI3C, and SWI3D, while the SWI3A/SWI3B pairs are more related to the branches of yeast and animal Swi3 sequences (Figure 1A). The conserved domain analyses showed that all Swi3-like proteins contain two characteristic domains: SWIRM and SANT domains (Figure 1B). Moreover, the SWIRM-assoc_1 (SWIRM-associated region 1) domain (Interpro: IPR032451), which was previously identified as the leucine zipper domain [20], was found in all members of Swi3-like proteins except for SlSWI3B (Figure 1B). Like the AtSWI3D and OsSWI3Ds, a ZnF_ZZ domain (Interpro: IPR000433), which was named because of the ability to bind two zinc ions [29], was also found in the N-terminus of SlSWI3D. In general, ZZ-type zinc finger domains contained 4–6 Cys residues that participate in zinc binding as well as protein-protein interaction [30].

### 2.2. Subcellular Localization of Tomato Swi3-Like Proteins

Bioinformatics analysis showed that SlSWI3 proteins exhibit various patterns of subcellular localization (Table 1). Nevertheless, a previous study demonstrated that *Arabidopsis* AtSWI3B localizes at the nucleus [21]. To determine the subcellular localization of tomato Swi3-like proteins, we performed in vivo targeting experiments in tobacco (*N. benthamiana*). To this end, *SlSWI3A*, *SlSWI3B SlSWI3C* and *SlSWI3D* were subcloned into the pEAQ-GFP vector to generate *35S*: *SlSWI3A*-*GFP*, *35S*: *SlSWI3B*-*GFP*, *35S*: *SlSWI3C*-*GFP* and 35S: *SlSWI3D*-*GFP* constructs, respectively. The constructs were introduced into leaf cells of tobacco by *A. tumefaciens* infiltration [31]. The fluorescence was visualized through a laser-scanning confocal microscope, and DAPI staining was used to visualize nuclei localization. We found that the *N. benthamiana* epidermal cells exhibited strong fluorescence in the nuclei (Figure 2, Appendix A), suggesting that the tomato SlSWI3-like proteins are nuclear proteins.

### 2.3. The Expression Patterns of Tomato Swi3-Like Genes

In order to explore the possible roles of *SlSWI3s*, we first analyzed their tissue and organ specific expression profiles from publicly available RNA-seq datasets [32]. As shown in Figure 3A, the transcripts of all *SlSWI3s* were ubiquitously expressed in all tissues and organs. *SlSWI3C* and *SlSWI3D* had similar expression profiles and were expressed mainly in roots (R) and fruits from 1 cm to 10-day post breaker (B + 10) stages (Figure 3A), suggesting that these genes may play redundant roles in root and fruit development. *SlSWI3B* showed a high expression in buds (B), flowers (F), roots (R) and B + 10 fruits, but its expression was low in other tissues. Compare with other *SlSWI3s*, the expression level of *SlSWI3A* was low in all tissues, especially in leaves (L) and B + 10 fruits (Figure 3A). Indeed, all *SlSWI3s* displayed low expression patterns in L compared with other tissues (Figure 3A).

Next, we investigated the expression pattern of *SlSWI3s* in response to environmental stimuli including hormones, salt, and cold by qRT-PCR. *SlSWI3A* and *SlSWI3B* were clearly induced by cold treatment, and *SlSWI3B* was also induced by ABA and salt treatments (Figure 3B). The transcript of *SlSWI3C* was strongly repressed by ABA, salt and SA treatment. Unlike *SlSWI3C*, the expression of *SlSWI3A* and *SlSWI3B* was not changed under SA treatment (Figure 3B). Interestingly, the transcript of *SlSWI3D* showed no difference under all treatments compared to control (Figure 3B). These results revealed that *SlSWI3A, SlSWI3B,* and *SlSWIC* may be involved in response to different environmental stimuli in tomato.

### 2.4. Members of SlSWI3s Interact with Other Proteins

In *Arabidopsis*, yeast-two-hybrid screens showed that AtSWI3A and AtSWI3B interact with the MADS-box transcriptional factors AGL18 (AGAMOUS-LIKE 18) and AGL73 [33]. In addition, AtSWI3A and AtSWI3B form homodimers and heterodimers and interact with BSH/SNF5 [20], while AtSWI3B and AtSWI3C form heterodimers and interact with the Snf2-like protein BRM (the ATPase of the SWI/SNF chromatin-remodeling complex) [18,34]. Thus, to explore whether SlSWI3s showed the similar interaction pattern, yeast-two-hybrid assays were performed. Like AtSWI3A and AtSWI3B, SlSWI3A, and SlSWI3B can also form heterodimers and homodimers (Figure 4A, Appendix A). In addition, SlSWI3B also interacted with SlSWI3C in yeast cells (Figure 4A). The interactions of SlSWI3s with the SlRIN and the Snf2-like protein SlCHR8 were also analyzed. Indeed, SlRIN is encoded by a member of the *SEPALLATA4* (*SEP4*) clade of MADS-box genes [35], which function as a master regulator of the ripening process in tomato. SlCHR8 displays high sequence homology with BRM in *Arabidopsis* [36], and overexpression of *SlCHR8* in tomato resulted in considerably compacter growth including significantly shorter roots and hypocotyls as well as reduced cotyledon and fruit size [37]. The yeast-two-hybrid assay showed that SlRIN interacted with SlSWI3A, SlSWI3B, and SlSWI3C, while SlCHR8 only interacted with SlSWI3B (Figure 4A, Appendix A). Intriguingly, the yeast-two-hybrid assay also indicated that SlSWI3C can interacte with *Arabidopsis* SWI3A, SWI3B, and SWI3D, respectively (Appendix A). Taken together, these results suggested that the function of SWI3s may be conserved in tomato and *Arabidopsis*.

To confirm their interaction in plant cells, SlSWI3A, SlSWI3B, SlSWI3C, SlRIN and SlCHR8 were fused with Nluc or Cluc and coexpressed in tobacco leaves and subjected for firefly luciferase complementation imaging assays. Consistent with the yeast-two-hybrid results, strong LUC activity was observed when Cluc-SlSWI3B and SlSWI3A-Nluc/or SlSWI3C-Nluc/or SlCHR8-Nluc were coexpressed (Figure 4B). Similar results were also observed when SlSWI3B-Nluc and CLuc-SlRIN were coexpressed in tobacco leaves (Figure 4B). Next, the interaction of SlSWI3B with SlRIN and SlCHR8 was examined by co-immunoprecipitation (Co-IP) assays. We transiently expressed SlSWI3B, SlRIN and SlCHR8 proteins in tobacco. The SlSWI3B fused with three FLAG tags (pHB-SlSWI3B) and SlRIN (or SlCHR8) fused with a GFP tag (pEAQ-SlRIN) constructs were co-transformed into tobacco epidermal cells by *Agrobacterium*-mediated infiltration assays. The anti-GFP antibody (GFP-Trap^®^_A beads) was used for immunoprecipitation, and the immunoprecipitated protein was then analyzed by western-blotting assays using an anti-Flag antibody. We showed that SlSWI3B-FLAG protein was co-immunoprecipitated by SlRIN-GFP or SlCHR8-GFP (Figure 4C). Collectively, these data supported that SlSWI3B interacts with SlSWI3A, SlSWI3C, SlCHR8, and SlRIN both in vivo and in vitro.

### 2.5. Ectopic Overexpression of SlSWI3C Enhances Leaf Growth

In *Arabidopsis*, overexpression of *SWI3C* frequently leads to an increase in rosette area [26]. To further analyze their functions, *SlSWI3A*, *SlSWI3B*, *SlSWI3C,* and *SlSWI3D* were ectopically expressed in *Arabidopsis* plants driven by the 35S promoter. The independent hygromycin-resistant T_1_ transformants were transferred into soil and grown in a greenhouse and self-pollinated to obtain segregated T_2_ progeny for genetic analysis. All overexpression lines showed a 3:1 segregation pattern for hygromycin resistance, and three homozygous lines were selected for further analysis. As for *SlSWI3C*, all these selected lines showed increased expression of *SlSWI3C* (Appendix A). Phenotypic analysis showed that the 21-day-old soil-grown seedlings of *35S:SlSWI3C 1*, *35S*:*SlSWI3C 2* and *35S*:*SlSWI3C 3* were much larger than Col-0 (Figure 5A). Consistent with this observation, *35S:SlSWI3C 1*, *35S*:*SlSWI3C 2* and *35S*:*SlSWI3C 3* plants had significantly larger rosette leaves (Figure 5B). In addition, the surface area of the 5^th^ rosette leaf of *35S:SlSWI3C 1*, *35S*:*SlSWI3C 2* and *35S*:*SlSWI3C 3* is 38.2, 46.04, and 46.08 mm^2^, respectively, which were significantly larger than that of Col-0 (Figure 5C). Consistent with the larger leaves, the fresh weights (FW) of up-ground parts of all *SlSWI3C* overexpressing plants was increased when compared with Col-0 plants (Appendix A). In contrast, transgenic *Arabidopsis* plants overexpression of *SlSWI3A, SlSWI3B,* and *SlSWI3D* showed no visible phenotypes including the rosette sizes compared with Col-0 (Appendix A, Appendix A).

Next, we determined the role of *SlSWI3C* in the expression of the genes related to leaf development. As shown in Figure 5D, the expression levels of *GROWTH REGULATING FACTOR 3* (*GRF3)*, *GRF5*, and *GRF6* were significantly increased in 21-day-old *35S:SlSWI3Cs* shoots grown in long-day conditions compared with Col-0 (Figure 5D). GRFs are comprised of nine members (GRF1 to GRF9) [38] and stimulate leaf cell proliferation, since overexpression of *GRF1*, *GRF2*, and *GRF5* enhances leaf growth and cell division [39,40]. Thus, the increased expression of *GRFs* in *35S:SlSWI3Cs* may be responsible for the larger leaf size. Intriguingly, *HB33*, a gene that is upregulated in *brm* mutants [26], was significantly downregulated in *35S:SlSWI3Cs* plants (Figure 5D). 

## 3. Discussion

As one of the subunit of CRCs, SWI3 plays key roles in different cellular process in eukaryotic cells. In yeast, the SWI–SNF assembly and ATP-dependent H2A–H2B displacement are controlled by Swi3p [11,12]. Human BAF170 (SMARCC2) and BAF155 (SMARCC1), the homologs of Swi3p, are involved in cancers since BAF170 and BAF155 mutations were found in gastric and colorectal cancers and small cell lung cancers, respectively [41,42]. In *Arabidopsis*, both *AtSWI3A* and *AtSWI3B* are essential for early embryonic development, whereas *AtSWI3C* and *AtSWI3D* affect different phases of vegetative and reproductive development [20]. In this study, we identified four SWI3-like proteins in tomato: SWI3A, SWI3B, SWI3C, and SWI3D (Table 1, Figure 1A). Intriguingly, there are 6 SWI3-like proteins in rice (Table 1, Figure 1A) and 7 SWI3-like proteins in maize [27], indicating more SWI3-like proteins present in monocot. The fact that more SWI3 homologs exist in monocot may suggest the functional diversification of SWI3 paralogs in monocot.

All SlSWI3s contain the typical SWIRM and SANT domains (Figure 1B). The SWIRM domain forms a compact helix-turn-helix (HTH)-related structure and mediates specific protein-protein interactions [15], while the SANT domain, which is characterized by its homology to the DNA binding domain of c-myb, functions as a histone tail binding module [17,43]. In addition to SWIRM and SANT domains, a SWIRM-assoc_1 domain was found in the C-terminus of SlSWI3s (Figure 1B). A leucine zipper domain located on the C-terminus of SWI3s [20] is involved in the interaction between SWI3s, since human BAF155 and BAF170 form heterodimers (BAF155/BAF170) or homodimers (BAF155/155 or BAF170/170) through the leucine zipper domain [44]. The sequences of the SWIRM-assoc_1 domain are larger than that of leucine zipper domain and the function of the SWIRM-assoc_1 domain is still unclear. Similar to the *Arabidopsis* AtSWI3B [21], all SlSWI3s are also localized in the nucleus (Figure 2), which is consistent with their roles as the subunits of CRCs.

The tissue-specific and stress-responsive expression patterns may be a useful way to explore gene functions. The expression profiles of *SlSWI3s* suggested that some *SlSWI3s* may play a similar role with their homologs in *Arabidopsis*. For instance, *SlSWI3C* is mainly expressed in roots and reproductive organs (Figure 3A), indicating that it functions in root and reproductive development [20]. Furthermore, functional divergence was also observed between *SlSWI3B* and its homolog *AtSWI3B*, since *SlSWI3B* is only poorly expressed in leaves (Figure 3A), whereas *AtSWI3B* is highly expressed in this organ [45], and knock-down of *AtSWI3B* results in an upward-curling leaf phenotype [24]. In addition, we found that the expression of *SlSWI3A* and *SlSWI3B* is induced by cold treatment, while *SlSWI3C* is strongly repressed by ABA, salt and SA treatment (Figure 3B). These data indicate that some SlCHRs may also respond to environmental stimuli.

In *Arabidopsis*, AtSWI3A and AtSWI3B can form homodimers and heterodimers, AtSWI3C can form heterodimers with both AtSWI3A and AtSWI3B, whereas AtSWI3D can form heterodimers only with AtSWI3B [18,20], suggesting their functional together in a complicated controlling network. In addition, AtSWI3B interacts with BSH/SNF5 and BRM, two core subunits of the SWI/SNF chromatin-remodeling complex [20,34]. Consistent with these data, our results also showed that SlSWI3A and SlSWI3B can form heterodimers and homodimers, whereas SlSWI3B and SlSWI3C can form heterodimers (Figure 4, Appendix A). Moreover, SlSWI3C can also form heterodimers with AtSWI3s (Appendix A), and interacts with SlCHR8, the *Arabidopsis* homolog of BRM. Collectively, these data suggested that CRCs in *Arabidopsis* and tomato may have similar functions. Nevertheless, the interaction between SlSWI3s (SlSWI3A, SlSWI3B, and SlSWI3C) with the MADS-box protein SlRIN (Figure 4, Appendix A) indicated that SlSWI3s may be involved in fruit ripening in tomato. 

Previous studies showed that *AtSWI3B* is essential for early embryonic development and ABA signaling [20,21], while *AtSWI3C* and *AtSWI3D* play important roles in vegetative and reproductive development [20] as well as GA signaling [25]. Mutations of *AtSWI3C* cause leaf curling and reduced fertility, while mutations of *AtSWI3D* lead to leaf curling, severe dwarfism, and alteration in the number and development of flower organs with complete male and female sterility [20]. Moreover, over-expression of *AtSWI3C* increases the leaf size due to increased cell numbers [26]. *AtSWI3B* is also involved in leaf development via regulating *IAMT1*, which encodes a carboxyl methyltransferase in the auxin metabolism [24]. Intriguingly, the maize *SWI3D*-like gene *ZmCHB101* knock-down lines also show curling leaves and impaired development in reproductive tissues, such as significant reduction of spikelet numbers and smaller and lighter ears compared to WT [27]. Collectively, these observations indicated that the physiological functions of SWI3s may be evolutionarily conserved in different plant species.

Like *AtSWI3C*, over-expression of *SlSWI3C* also resulted in the increased leaf size in transgenic *Arabidopsis* (Figure 5A–C). The expression of *GRF3*, *GRF5,* and *GRF6* was significantly increased in *SlSWI3C* overexpressing seedlings (Figure 5D). These data suggested that the bigger leaf size of over-expression *SlSWI3C* transgenic *Arabidopsis* may be caused by the increased expression of *GRFs*, since overexpression of *GRF1*, *GRF2*, and *GRF5* enhances leaf growth and cell division [39,40]. Nevertheless, transgenic *Arabidopsis* seedlings overexpression of *SlSWI3A*, *SlSWI3B,* and *SlSWI3D* showed no difference in the rosette sizes compared with Col-0 (Appendix A), suggesting that SlSWI3C may function differently from other SISWI3 proteins. Taken together, our results indicated that the functions of SWI3C may, at least in some developmental processes, be conserved in tomato and *Arabidospis*. Further in-depth analysis using transgenic tomato is required to investigate the functions of SlSWI3s in tomato development.

## 4. Materials and Methods

### 4.1. Plant Materials and Growth Conditions

*Solanum lycopersicum* cultivar “Heinz 1706” was used in this study. Surface-sterilized tomato seeds were grown in the Murashige and Skoog (MS) medium with 1.5% sucrose and 0.8% agar for 14 days in a controlled environment greenhouse with a long photoperiod (16 h light/8 h dark) at 23 ± 1 °C.

For hormone and stress treatments, 14-day-old “Heinz 1706” seedlings grown in the MS medium were transferred to the liquid MS medium containing SA (5 mM), ABA (100 μM), and NaCl (200 mM) for 2 h, respectively. For cold stress test, the plates were transferred to a 4 °C growth cabinet for 2 h. After treatment, the seedlings were harvested and immediately frozen in liquid nitrogen for further gene expression. The seedlings without treatment (seedlings grown in the MS medium) were used as control. For each treatment, about 10 seedlings were used for RNA extraction.

### 4.2. Identification of Tomato Swi3-Like Genes and Phylogenetic Tree Construction

The AtSWI3A, AtSWI3B, AtSWI3C and AtSWI3D sequences of *Arabidopsis thaliana* and yeast ScSWI3 were used to perform a search in the *Solamum lycopersicum* genome using the BLASTP program in the SGN (http://solgenomics.net/). Then, the candidates of Swi3-like proteins were confirmed using the HMMER-based SMART (http://smart.embl-heidelberg.de/) and Pfam (http://pfam.xfam.org/) programs. The domain architecture was drawn using DOG2.0 software [46].

The Swi3-like protein sequences from tomato, *Arabidopsis*, rice (*O. sativa*), yeast (*S. cerevisiae*), fruit fly (*D. melanogaster*) and human (*H. sapiens*) were aligned with ClustalW, and the alignment was imported in MEGA5.2 for phylogenetic generation using Neighbor-Joining method [47].

### 4.3. Subcellular Localization Assays

The subcellular localizations of SlSWI3s were first predicted using the WoLFPSORT (http://www.genscript.com/psort/wolf_psort.html), Plant-mPLoc (http://www.csbio.sjtu.edu.cn/bioinf/plant-multi/#) and Euk-mPLoc 2.0 (http://www.csbio.sjtu.edu.cn/bioinf/euk-multi-2/) programs, and confirmed by GFP-tagged transient expression assays in tobacco (*Nicotiana benthamiana*) leaves [31]. The full length CDS of *SlSWI3s* were subcloned into the pEAQ-GFP vector (the C-terminus of SlSWI3s was fused with GFP) [48], and all the constructs were subsequently transformed into *Agrobacterium tumefaciens* strain GV3103. After grown in LB medium at 28 °C overnight, bacterial suspensions were infiltrated into young but fully expanded leaves of tobacco using a needleless syringe. After infiltration, plants were immediately covered with plastic bags and placed at 23 °C for 48 h before bag removal. The distribution of the fusion protein was determined using a confocal fluorescence microscope. To locate the fluorescent proteins in nuclei, the tobacco leaves were infiltrated with PBS containing 4′, 6-diamidino-2-phenylindole (DAPI, 1μg/mL). Three independent experiments were repeated to confirm results.

### 4.4. Yeast-Two-Hybrid Assay

Yeast-two-hybrid assays were performed according to the manufacturer’s instructions for the Matchmaker GAL4-based two-hybrid system 3 (Clontech, Takara, Dalian, China). *SlSWI3s*, *SlCHR8* and *SlRIN* were cloned into the pGBKT7 or pGADT7 vectors. Protein interactions were tested by stringent (SD/-Leu/-Trp/-His/-Ade) selection supplied with β-galactosidase activity measurement (Clontech, Takara, Dalian, China). Three independent experiments were repeated to confirm results.

### 4.5. LCI Assay

Luciferase complementation imaging (LCI) assays were performed as described previously [49]. The CDS of *SlSWI3A*, *SlSWI3B*, *SlSWI3C*, *SlCHR8,* and *SlRIN* was cloned into either pCAMBIA-Nluc or pCAMBIA-Cluc. All the constructs were transformed into *A. tumefaciens* strain GV3103. An equal volume of *A. tumefaciens* harboring pCAMBIA-NLuc and pCAMBIA-CLuc (or their derivative constructs) was mixed to a final concentration of OD_600_ = 1.0. Four different combinations of *A. tumefaciens* were infiltrated into four different positions at the same leaves of tobacco. Plants were placed in 23 °C and allowed to recover for 60 h. A low-light cooled CCD imaging apparatus (NightOWL II LB983 with indiGO software) was used to capture the LUC image. Three independent experiments were repeated to confirm results.

### 4.6. Co-Immunoprecipitation (Co-IP) Assays

Co-IP assays were performed as described previously [50]. Two days after infiltration, tobacco leaves were harvested and ground to a fine powder in liquid nitrogen. Proteins were extracted in an extraction buffer (50 mM Tris-HCl, pH 7.4, 150 mM NaCl, 2 mM MgCl_2_, 1 mM DTT, 20% glycerol, and 1% NP-40) containing protease inhibitor cocktail (Roche, Shanghai, China). Cell debris was pelleted by centrifugation at 14,000× *g* for 20 min. The supernatant was incubated with 30 μL of GFP-Trap^®^_A beads (Chromo Tek, Martinsried, Germany) at 4 °C for 4 h, then the beads were centrifuged and washed six times with a washing buffer (50 mM Tris-HCl, pH 7.4, 150 mM NaCl, 2 mM MgCl_2_, 1mM DTT, 10% glycerol, and 1% NP-40). Proteins were eluted with 40 μL of 2× loading buffer and analyzed by western blotting using anti-GFP (Roche, Shanghai, China) and anti-Flag antibodies (LifeTein, Beijing, China).

### 4.7. RNA Extraction and Expression Analyses

Total RNA was isolated using Trizol Reagent (Invitrogen, Shanghai, China) according to the manufacturer’s protocol. cDNAs were synthesized from 2 μg of total RNA using the TransScript^™^ One-Step gDNA Removal and cDNA Synthesis Supermix kit (TransGenBiotech, Guangzhou, China). Real-Time PCR was performed with iTaq^™^ Universal SYBR^®^ Green Supermix (BIO-RAD, Shanghai, China) using ABI7500 Fast Real-Time PCR system. The gene-specific primers for real-time PCR were designed by PrimerQuest Tool (https://sg.idtdna.com/Primerquest/Home/Index) and listed in Appendix A. The *Arabidopsis ACTIN2* (*AT3G18780*) and tomato *ACTIN* (Solyc03g078400) were used as a reference gene. Three independent sets of biological replicates were conducted with two technical replicates to confirm results.

For tissue and organ specific expression profiles, the expression data of tomato *SlSWI3s* were extracted from publicly available RNA-seq datasets from the Tomato Genome Consortium [32] and visualized with Matrix2PNG (https://matrix2png.msl.ubc.ca/bin/matrix2png.cgi) [51]. The publicly available RNA-seq data were obtained from transcriptome sequencing using three-week-old sand-grown seedlings, roots, leaves, buds (unopened flower buds), and flowers (fully open flowers) as well as fruits (at 1 cm, 2 cm, and 3 cm), MG (mature green), breaker (Br, early ripening), and 10-day post-breaker (B + 10, red ripe) stages of tomato “Heinz 1706” [32]. The expression data of *SlSWI3s* were normalized to have mean zero and variance one before producing the heat maps.

### 4.8. Generation of Transgenic Plants

To construct the *SlSWI3C* and *SlSWI3D* overexpression vectors, the CDS without a stop codon of *SlSWI3C* and *SlSWI3D* were cloned into the plasmid pHB-flag containing the 35S promoter [52]. The constructed plasmids were separately transformed into the *A. tumefaciens strain* GV3103 by the heat shock method. The bacteria carrying different constructs were used to transform WT (Col-0) plants via floral dip transformation.

Leaf areas were measured with ImageJ (http://rsb.info.nih.gov/ij/) using three-week-old seedlings grown in soil (the same stage seedlings were also for RNA isolation) after dissection of individual leaves. Rosette areas were calculated as the sum of the individual leaf areas. Three independent sets of biological replicates were conducted to confirm results.

## 5. Conclusions

In this study, four Swi3-like proteins, SlSWI3A, SlSWI3B, SlSWI3C, and SlSWI3D, were identified from tomato. All SlSWI3s contain two characteristic domains: SWIRM and SANT domains, and are localized in the nucleus. All *SlSWI3s* are ubiquitously expressed in all tissues and organs, and *SlSWI3A* and *SlSWI3B* can be induced by cold treatment. In addition, SlSWI3B forms homodimers with itself and heterodimers with SlSWI3A and SlSWI3C. SlSWI3B also interacts with SlRIN and SlCHR8, two proteins associated with tomato reproductive development, indicating that SlSWI3B may be involved in gene regulation in reproductive development. Furthermore, over-expression of *SlSWI3C* increases the leaf size in transgenic *Arabidopsis* with increased expression of *GRF3*, *GRF5* and *GRF6*.

## Figures and Tables

**Figure 1 ijms-20-05121-f001:**
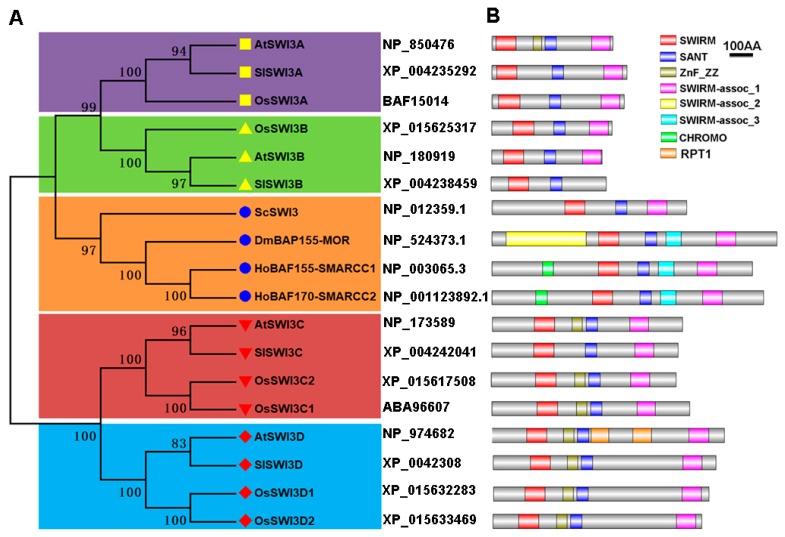
Phylogenetic tree and domain architecture of SlSWI3s in tomato. (**A**) Phylogenetic tree of Swi3-like proteins in plants and animal. Neighbor-joining (NJ) phylogenetic tree for SWI3s in *S. cerevisiae* (Sc), *D. melanogaster* (Dm), *H. sapiens* (Ho), *A. thaliana* (At), and *S. lycopersicum* (Sl). The groups of homologous genes identified, bootstrap values and accession numbers are shown. The reliability of branching was assessed by the bootstrap resampling method using 1000 bootstrap replicates. (**B**) Domain architecture of the Swi3-like proteins was drawn by DOG2.0. according to analysis by SMART and PFAM searches. The location of domains is shown by different color as indicated. The scale represents the length of the protein and all proteins are displayed in proportion. The proteins belonging to each family are grouped together.

**Figure 2 ijms-20-05121-f002:**
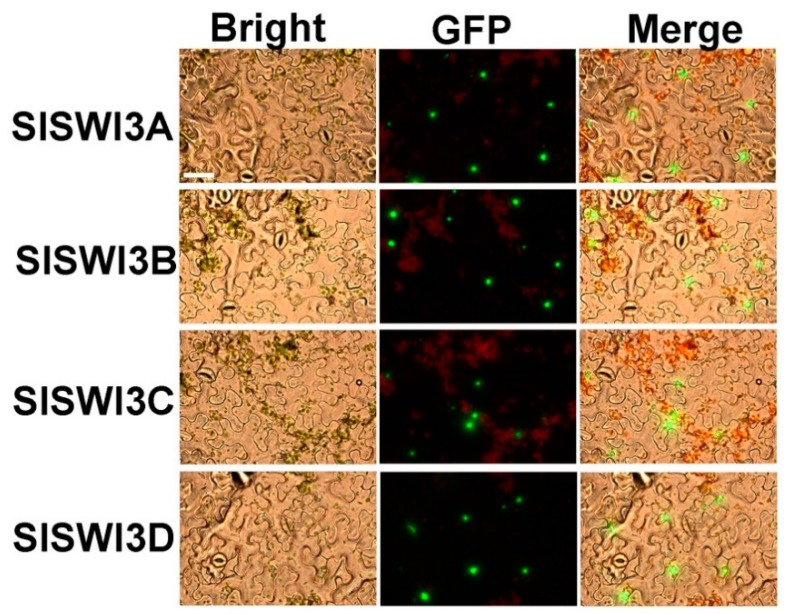
Subcellular localization of SlSWI3s. *A. tumefaciens* strain GV3103 harboring *SlSWI3s-GFP* constructs were infiltrated into young but fully expanded leaves of tobacco. After growth at 23 °C for 48 h, the epidermis of tobacco leaves were used to determine the distribution of the fusion protein using a confocal fluorescence microscope. The MADS-box transcriptional factor SlRIN was used as a positive control for localization of a nuclear protein. Bars = 50 μm. Similar results were obtained in three independent experiments, and one representative result was shown.

**Figure 3 ijms-20-05121-f003:**
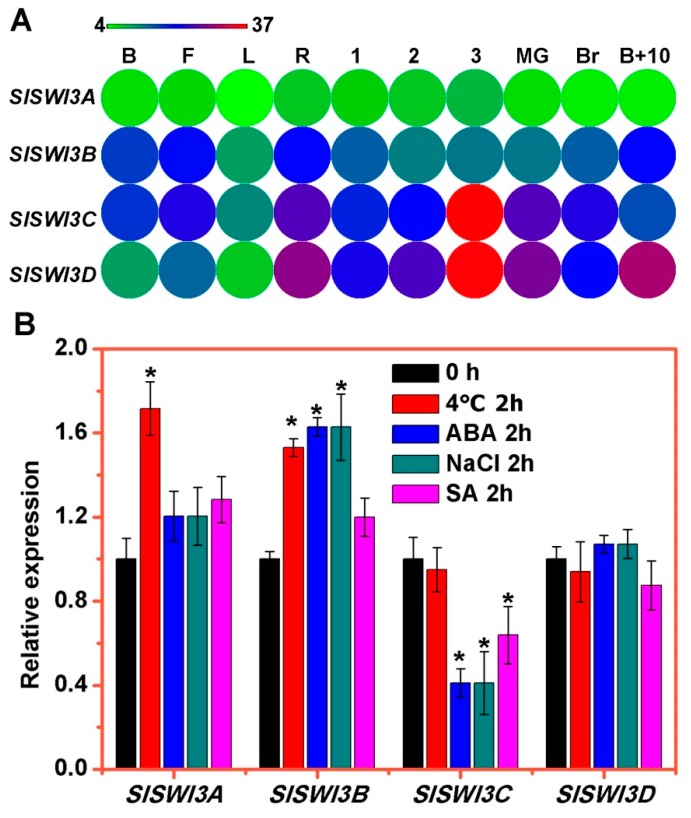
The expression patterns *SlSWI3s*. (**A**) Tissue-specific expression patterns of *SlSWI3s*. Heat map of RNA-seq expression data from bud (B), flower (F), leaf (L), root (R), 1 cm_fruit (1), 2cm_fruit (2), 3cm_fruit (3), mature green fruit (MG), berry at breaker stage (Br), and berry ten days after breaking (B + 10). The expression values are measured as reads per kilobase of the exon model per million mapped reads (RPKM). (**B**) Expression profiles of *SlSWI3s* under responding to hormones, salt, and cold tested by qRT-PCR. Seedlings of two-week-old plate-cultured plants were treated with SA (2 mM), ABA (100 μM), NaCl (200 mM), and cold (4 °C) for 2 h and collected for total RNA isolation. qRT-PCR was amplified using gene-specific primers. The tomato *ACTIN* (*Solyc03g078400*) was used as an internal control. Error bars indicate the SE. Asterisks indicate significant difference from wild-type plants (*p* < 0.05, Student’s *t* test). The data are representative from three independent experiments.

**Figure 4 ijms-20-05121-f004:**
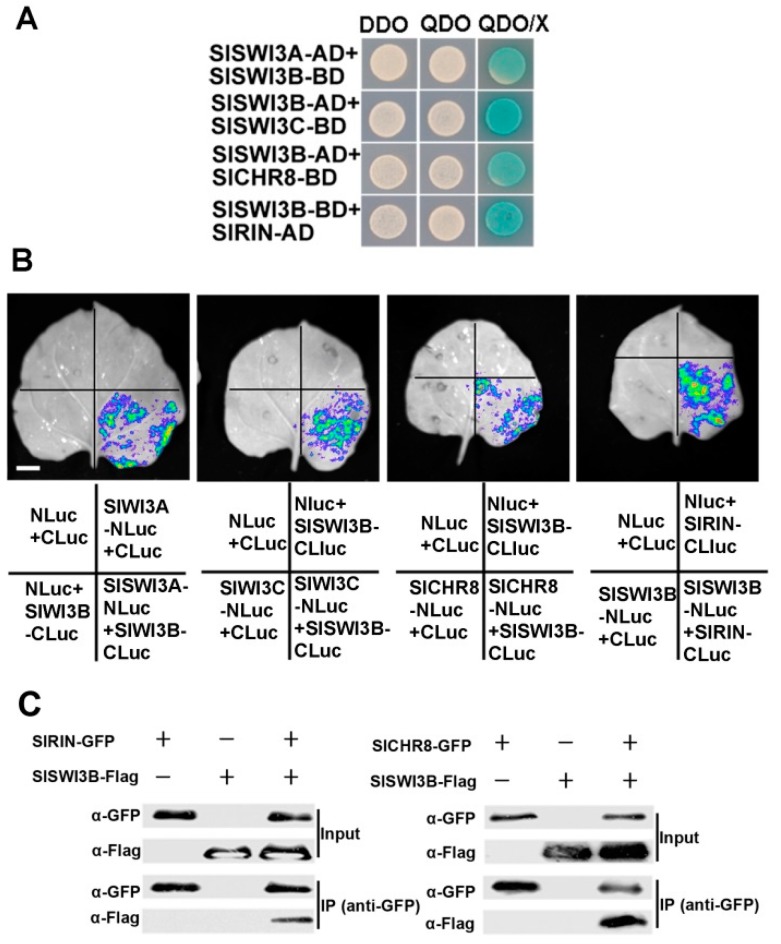
SlSWI3s interacted with SlSWI3A, SlSWI3C SlRIN, and SlCHR8 proteins. (**A**) SlSWI3B interacted with SlSWI3A, SlSWI3C SlRIN and SlCHR8 in yeast-two-hybrid assays. *SlSWI3B* was either cloned into pGBKT7 or pGADT7 vectors, whereas *SlSWI3A*, *SlSWI3C* and *SlCHR8* were cloned into pGBKT7 vector, and *SlRIN* was cloned into pGADT7 vectors, respectively. Different constructs were cotransformed into the yeast strain AH109. The transformants were grown on the selective minimal medium without Leu and Trp (DDO). The transformants were also plated on QDO or QDO/X to test for possible interaction. QDO, SD/-Leu/-Trp/-His/-Ade. X, x-a-gal. The same results were obtained in three independent experiments, and one representative result was shown. (**B**) SlSWI3B interacted with SlSWI3A, SlSWI3C SlRIN and SlCHR8 by LCI assay. The *A. tumefaciens* carrying the indicated construct pairs were injected into tobacco leaves, and the luciferase activities were measured 2 d after injection. Similar results were obtained in three independent experiments, and one representative result was shown. Bars = 0.5 cm. (**C**) SlSWI3B interacted with SlRIN and SlCHR8 by Co-IP assay.The *SlSWI3B* and *SlRIN* (or *SlCHR8*) were subcloned into the pHB (Flag tag) and pEAQ-GFP (GFP tag) vector, respectively.These constructs were co-transformed into tobacco cells by Agrobacterium mediated infiltration. Transiently expressed SlSWI3B-Flag and SlRIN-GFP (or SlCHR8-GFP) were immunoprecipitated with an anti-GFP antibody, and then detected by western-blotting assay with an anti-Flag antibody.

**Figure 5 ijms-20-05121-f005:**
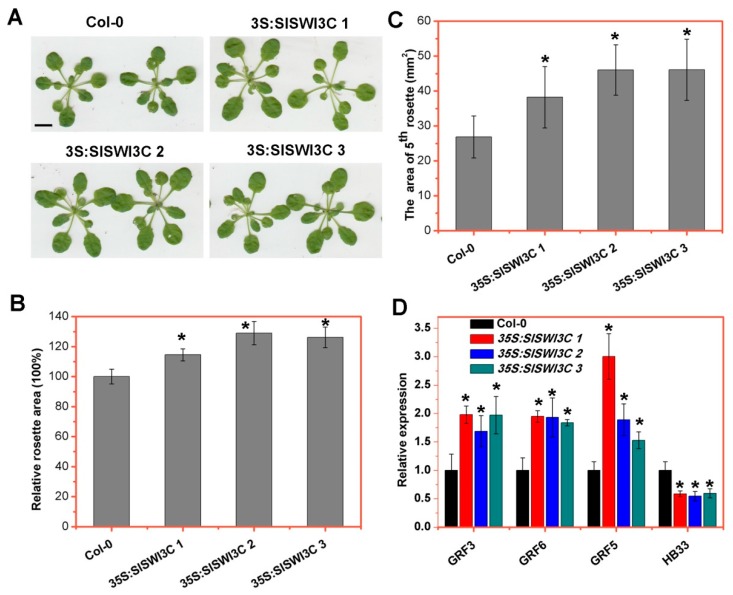
Overexpression of SlSWI3C enhances leaf growth. (**A**) Rosettes of 21-d-old Col-0 and 35S:SlSWI3C lines showing enhanced leaf growth. Three experiments were repeated with similar results, and one representative result was shown. Bar = 0.5 cm. (**B**, **C**) Total (**B**) and 5th rosette (**C**) area calculated using ImageJ (http://rsb.info.nih.gov/ij/) software. The individual leaf from 21-d-old Col-0 and *35S:SlSWI3Cs* plants was photographed and calculated using ImageJ. Error bars are SE (*n* = 15). Asterisks indicate significant difference from the Col-0 (*p* < 0.05, Student’s *t* test). The data are representative from three independent experiments. (**D**) *SlSWI3C* promotes the expression of *GRFs* in *Arabidopsis*. Expression levels determined by qRT-PCR in 21-d-old Col-0 and *35S:SlSWI3Cs* rosettes grown in long-day (16 h light/8 h dark) conditions. qRT-PCR was amplified using gene-specific primers. The *ACTIN2* (*AT3G18780*) was used as an internal control. Error bars are SE. Asterisks indicate significant difference from wild-type plants (*p* < 0.05, Student’s *t* test). The data are representative from three independent experiments.

**Table 1 ijms-20-05121-t001:** The SWI3-like proteins in Tomato, *Arabidopsis* and rice.

SWI3 Gene Family	Gene Name	Gene Code	Accession Number ^a^	ORF Length ^b^	Protein Length	Localization ^c^	Number of Exons	Nuclear Localization Signal ^d^
SWI3A	*SlSWI3A*	Solyc03g097450	XP_004235292	1772	574	nucl, cyto	7	Yes
	*AtSWI3A*	AT2G47620	NP_850476	1539	513	nucl, cyto	7	Yes
	*OsSWI3A*	LOC_Os04g40420	BAF15014	1683	561	nucl, cyto, plas	7	Yes
SWI3B	*SlSWI3B*	Solyc04g082760	XP_004238459	1470	490	nucl, cyto	6	Yes
	*AtSWI3B*	AT2G33610	NP_180919	1410	469	nucl, cyto	6	Yes
	*OsSWI3B*	LOC_Os02g10060	XP_015625317	1536	512	nucl	6	Yes
SWI3C	*SlSWI3C*	Solyc06g060120	XP_004242041	2376	792	nucl, chlo, mito	8	Yes
	*AtSWI3C*	AT1G21700	NP_173589	2424	807	nucl, chlo, plas	9	Yes
	*OsSWI3C1*	LOC_Os12g07730	ABA96607	2520	840	nucl, chlo, mito	9	Yes
	*OsSWI3C2*	LOC_Os11g08080	XP_015617508	2355	785	nucl, chlo, mito	9	Yes
SWI3D	*SlSWI3D*	Solyc01g109510	XP_004230866	2838	946	nucl	6	Yes
	*AtSWI3D*	AT4G34430	NP_974682	2961	986	nucl	7	Yes
	*OsSWI3D1*	LOC_Os03g51220	XP_015632283	2745	915	nucl, cyto	8	Yes
	*OsSWI3D2*	LOC_Os04g01970	XP_015633469	2661	887	nucl, chlo, mito	7	Yes

^a^ Accession numbers of full-length protein sequence available at NCBI (https://www.ncbi.nlm.nih.gov/). ^b^ Length of open reading frame (number of basepair). ^c^ Localization of tomato SWI3-like proteins supported by WoLFPSORT (http://www.genscript.com/psort/wolf_psort.html), Plant-mPLoc (http://www.csbio.sjtu.edu.cn/bioinf/plant-multi/#) and Euk-mPLoc 2.0 (http://www.csbio.sjtu.edu.cn/bioinf/euk-multi-2/). All SWI3-like proteins of plants was localized in the nucleus using Plant-mPLoc and Euk-mPLoc 2.0 programs. cyto, cytoplasm; nucl, nucleus; Plas, plasmamembrane; mito, mitochondrion; chlo, chloroplast. ^d^ Nuclear localization signal prediction based on frequent pattern mining and linear motif scoring using SeqNLS (http://mleg.cse.sc.edu/seqNLS/) and NucPred (https://nucpred.bioinfo.se/cgi-bin/single.cgi) programs.

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
