# Peer review of "Identification and Characterization of Tomato SWI3-Like Proteins: Overexpression of SlSWIC Increases the Leaf Size in Transgenic Arabidopsis"

_ijms, 2019, doi:10.3390/ijms20205121_

Round 1
Reviewer 1 Report
In this manuscript, the authors identified and analyzed four tomato SWI3 genes. Some of data need to be further verified or added, for examples: interactions and the expression. More importantly, the authors try to study the functions of tomato SWI3 genes, however, didn’t perform much useful work in the tomato. I feel the authors only performed several experiments: the interactions (in tobacco) and developments (in Arabidopsis), following the known Arabidopsis side. The novelty is low. The authors should deepen their research target and add the novelty in tomato side, such as at least showing how the knocking-down/out of tomato SlSWI3 genes influences the tomato development? or the interaction mechanisms and functions of SlSWI3 proteins and SlRIN and SlCHR8 in tomato? or the function research of SWI3 proteins against stress or hormones based on the expression analysis upon stress and hormones. The transgene of tomato may be slow, however, the authors can use the VIGS on tomato seedlings for preliminarily repaid functional research.
Other concerns:
The authors should show the accession numbers of SWI3 proteins from other plant and animal species in the Figure 1 tree.
In the figure 2, if the authors want to use SlRIN as a control of nuclear localization, the SlRIN protein should be fused to such as mCherry or CFP to show the co-localization of SlRIN and SlSWI3 proteins.
Line 219: just showing the interactions may be conserved between Arabidopsis and tomato.
In figure 4, the authors need to use Co-IP to verify their interactions, at least of SlSWI3 proteins and SlRIN and SlCHR8.
The authors should present the overexpression analysis results of SWI3B, C and D in overexpression lines.
The authors can perform the expression of SlSWI3C in the Arabidopsis AtSWI3C null mutant, to verify the conserved function of SWI3C in the plant development.
Line 118 and 190: qRT-PCR
Line 420 GV3103 by
Author Response
In this manuscript, the authors identified and analyzed four tomato SWI3 genes. Some of data need to be further verified or added, for examples: interactions and the expression. More importantly, the authors try to study the functions of tomato SWI3 genes, however, didn’t perform much useful work in the tomato. I feel the authors only performed several experiments: the interactions (in tobacco) and developments (in Arabidopsis), following the known Arabidopsis side. The novelty is low. The authors should deepen their research target and add the novelty in tomato side, such as at least showing how the knocking-down/out of tomato SlSWI3 genes influences the tomato development? or the interaction mechanisms and functions of SlSWI3 proteins and SlRIN and SlCHR8 in tomato? or the function research of SWI3 proteins against stress or hormones based on the expression analysis upon stress and hormones. The transgene of tomato may be slow, however, the authors can use the VIGS on tomato seedlings for preliminarily repaid functional research.
Response: Thank you for your suggestion. Indeed, we generated the SlSWI3C knocked-down transgenic tomato using PTG/Cas9 systems developed by Wang et al. (Wang et al., Optimized paired-sgRNA/Cas9 cloning and expression cassette triggers high-efficiency multiplex genome editing in kiwifruit, 2018, Plant Biotechnology Journal). However, no visible phenotypes were observed in these plants, suggesting that there is functional redundancy among the tomato SWI3 genes. To further investigate the functions of SlSWI3s in tomato development, we are in the progress to obtain the transgenic tomato with SlSWI3B/C overexpressed.
Other concerns:
The authors should show the accession numbers of SWI3 proteins from other plant and animal species in the Figure 1 tree.
Response: Thank you for your suggestion. We added the accession numbers in Figure 1.
In the figure 2, if the authors want to use SlRIN as a control of nuclear localization, the SlRIN protein should be fused to such as mCherry or CFP to show the co-localization of SlRIN and SlSWI3 proteins.
Response: Thank you for your suggestion. We removed the results of SlRIN (Figure 2).
Line 219: just showing the interactions may be conserved between Arabidopsis and tomato.
Response: Thank you for your suggestion. We changed this sentence in the revised manuscript (line 237-239).
In figure 4, the authors need to use Co-IP to verify their interactions, at least of SlSWI3 proteins and SlRIN and SlCHR8.
Response: Thank you for your suggestion. We performed Co-IP assays and data was shown in the revised manuscript (Figure 4 C, line 247-256).
The authors should present the overexpression analysis results of SWI3B, C and D in overexpression lines.
Response: Thank you for your suggestion. The expression of SlSWI3A, SlSWI3B and SlSWI3D was added in the revised manuscript (Supplemental Figure 4).
The authors can perform the expression of SlSWI3C in the Arabidopsis AtSWI3C null mutant, to verify the conserved function of SWI3C in the plant development.
Response: Thank you for your suggestion. Because the homozygous of swi3c plants showed greatly reduced fertility, thus it is difficult to obtain SlSWI3C overexpressing plants in swi3c background. Like AtSWI3C, overexpression of SlSWI3C also caused larger leaves in Arabidopsis indicating its conserved function in leaves development.
Line 118 and 190: qRT-PCR
Response: Thank you for your suggestion. We changed RT-PCR as qRT-PCR in the revised manuscript.
Line 420 GV3103 by
Response: Thank you for your suggestion. We added a space in the revised manuscript (line 468).
Reviewer 2 Report
In “Identification and characterization of tomato SWI3-like proteins: overexpression of SlSWIC increases the leaf size in transgenic Arabidopsis”, the authors following a complete molecular characterization provided convincing functional evidences for a roles of tomato SWI3-like proteins in the control of leaf size.
The paper is well written, the figures are of good quality and comprehensive with sufficient details provided in their legends.
The results are well presented and the conclusion are in my opinion logical and easy to follow.
The discussion is of good quality.
My only very minor concern is about biomass; indeed, the authors have evaluated leaf area (increased leaf area as a consequence of tomato SWI3-like over-expression), but what about biomass of these leaves. Leaf surface could increase but leaves could be finer and therefore with no change in biomass but changes in the global leaf structure. If it’s not possible I suggest to the authors to at least considerate this point, if relevant. If not, please provide me some arguments.
Author Response
In “Identification and characterization of tomato SWI3-like proteins: overexpression of SlSWIC increases the leaf size in transgenic Arabidopsis”, the authors following a complete molecular characterization provided convincing functional evidences for a roles of tomato SWI3-like proteins in the control of leaf size.
The paper is well written, the figures are of good quality and comprehensive with sufficient details provided in their legends.
The results are well presented and the conclusion are in my opinion logical and easy to follow.
The discussion is of good quality.
My only very minor concern is about biomass; indeed, the authors have evaluated leaf area (increased leaf area as a consequence of tomato SWI3-like over-expression), but what about biomass of these leaves. Leaf surface could increase but leaves could be finer and therefore with no change in biomass but changes in the global leaf structure. If it’s not possible I suggest to the authors to at least considerate this point, if relevant. If not, please provide me some arguments.
Response: Thank you for your suggestion. We added new contents in the revised manuscript (Supplemental Figure 5, line 274-276).
Reviewer 3 Report
The paper is well organized, with a clear aims and an introduction that gives a solid background for understanding the study of the authors. Results seem to support the discussion, which is additionally confronted with other key-information by different authors.
Author Response
The paper is well organized, with a clear aims and an introduction that gives a solid background for understanding the study of the authors. Results seem to support the discussion, which is additionally confronted with other key-information by different authors.
Response: Thank you.
Round 2
Reviewer 1 Report
Addressed my concerns.